# Association of Genetic Markers with the Risk of Early-Onset Breast Cancer in Kazakh Women

**DOI:** 10.3390/genes15010108

**Published:** 2024-01-17

**Authors:** Liliya Skvortsova, Saltanat Abdikerim, Kanagat Yergali, Natalya Mit, Anastassiya Perfilyeva, Nazgul Omarbayeva, Aigul Zhunussova, Zulfiya Kachiyeva, Tolkyn Sadykova, Bakhytzhan Bekmanov, Dilyara Kaidarova, Leyla Djansugurova, Gulnur Zhunussova

**Affiliations:** 1Laboratory of Molecular Genetics, Institute of Genetics and Physiology, Almaty 050060, Kazakhstan; lilia_555@rambler.ru (L.S.); abdikerimse@gmail.com (S.A.); ergali.0394@mail.ru (K.Y.); nata-mit@yandex.kz (N.M.); nastyaper2009@mail.ru (A.P.); aigul.s.zhunussova@gmail.com (A.Z.); bobekman@rambler.ru (B.B.); leylad@mail.ru (L.D.); 2Department of Molecular Biology and Genetics, Al-Farabi Kazakh National University, Almaty 050040, Kazakhstan; 3Breast Cancer Department, Kazakh Institute of Oncology and Radiology, Almaty 050060, Kazakhstan; nomarbayeva1@gmail.com (N.O.); sadykova.tolkyn@mail.ru (T.S.); dilyara.kaidarova@gmail.com (D.K.); 4Oncology Department, Asfendiyarov Kazakh National Medical University, Almaty 050012, Kazakhstan; 5Research Institute of Applied and Fundamental Medicine, Asfendiyarov Kazakh National Medical University, Almaty 050012, Kazakhstan; kachieva@gmail.com

**Keywords:** early-onset breast cancer, microarray-based SNP genotyping, genome-wide association study, genetic variations, genetic markers, Kazakh population

## Abstract

Breast cancer is a global health problem. It is an age-dependent disease, but cases of early-onset breast cancer (eBC) are gradually increasing. There are many unresolved questions regarding eBC risk factors, mechanisms of development and screening. Only 10% of eBC cases are due to mutations in the *BRCA1/BRCA2* genes, and 90% have a more complex genetic background. This poses a significant challenge to timely cancer detection in young women and highlights the need for research and awareness. Therefore, identifying genetic risk factors for eBC is essential to solving these problems. This study represents an association analysis of 144 eBC cases and 163 control participants to identify genetic markers associated with eBC risks in Kazakh women. We performed a two-stage approach in association analysis to assess genetic predisposition to eBC. First-stage genome-wide association analysis revealed two risk intronic loci in the *CHI3L2* gene (*p* = 5.2 × 10^−6^) and *MGAT5* gene (*p* = 8.4 × 10^−6^). Second-stage exonic polymorphisms haplotype analysis showed significant risks for seven haplotypes (*p* < 9.4 × 10^−4^). These results point to the importance of studying medium- and low-penetrant genetic markers in their haplotype combinations for a detailed understanding of the role of detected genetic markers in eBC development and prediction.

## 1. Introduction

Breast cancer (BC) is a significant health concern in Kazakhstan, as it is in many other countries around the world. According to the World Health Organization (WHO), BC is the most common cancer among women in Kazakhstan, accounting for approximately 22% of all cancers in females. Several factors contribute to the high incidence of BC in Kazakhstan. These include limited access to preventative and diagnostic services, lack of awareness about breast cancer screening and early detection, and inadequate healthcare infrastructure in certain regions of the country. To address these issues, the Kazakhstani government, in collaboration with international organizations, has implemented various initiatives to improve breast cancer prevention, screening, and treatment. These initiatives include public awareness campaigns, training of healthcare providers, and the establishment of breast cancer screening programs [1,2,3].

However, challenges remain in combating breast cancer in Kazakhstan. Limited resources, socioeconomic disparities, and cultural factors can hinder the effectiveness of these initiatives. Additionally, there is a need for further research and data collection to understand better the specific risk factors and patterns of breast cancer in the country.

Genome-Wide Association Study (GWAS) is an approach typically employed to identify common genetic variants associated with specific diseases or traits. It involves comparing the genomes of individuals with and without the disease or trait to pinpoint specific genetic markers [4]. In the case of breast cancer, GWAS has been instrumental in revealing numerous genetic variants associated with an increased risk of developing the disease. Over the years, several large-scale GWASs have identified numerous genetic variants linked to breast cancer susceptibility. Some of these variants are found in genes involved in DNA repair, estrogen metabolism and cell growth regulation.

The information obtained from GWAS has expanded our understanding of the genetic basis of breast cancer, contributing to the development of more accurate risk prediction models [5,6]. This allows individuals with a higher genetic risk for breast cancer to undertake proactive measures for early detection and prevention, such as increased surveillance and preventive surgeries.

GWAS analyzes high-frequency genetic markers, which are widespread in different populations (minor allele frequency more than 3% (MAF > 0.03)), in contrast to rare but pathologically significant mutations (mutations in the *BRCA1_c.5266dup*, *185delAG*, *BRCA2_6174delT* genes [7]). As a rule, these genetic markers or polymorphisms have medium or low penetrance of phenotype traits. Often, phenotypic manifestations of these polymorphisms are visible only at the molecular level (by experimental methods or prediction structure tools). They do not affect the total destruction of protein function, but milder modifications of its expression (promoter polymorphisms), transcriptions/translations (splice-site polymorphisms), mRNA bioavailability (synonymous variants), and structures that potentially influence protein interaction with targets (missense variants). These polymorphisms, apparently, have a longer time for accumulating initial (predisposing) events of cancer processes and are more dependent on the influence of external modifying factors. Therefore, GWAS shows that most of the identified high-frequency polymorphisms are not extrapolated to other populations with different environmental, cultural, socioeconomic, and lifestyle features. Undoubtedly, they are essential for the functioning of an organism. However, their contribution to phenotypic characteristics is hard to detect since their functional significance manifested in interaction with other factors (internal and external).

The genetic structure of the ethnic Kazakh women population is insufficiently studied. Some studies devoted mainly to *BRCA1* and *BRCA2* genes and other BC-associated genes have been conducted in various regions of Kazakhstan [8,9,10,11]. Moreover, there are no data on early-onset BC-associated SNP markers. This study presents for the first time the results of the GWAS that allow us to identify SNP markers previously uncharacterized for this group of patients.

Given the above, determining the general population and individual risk of developing breast cancer becomes a difficult task. Risk assessment should be complex and consider multiple interactions of various factors currently under research.

In this regard, in our study, we assessed the significance not only of separate genetic markers but their combinations (haplotypes) in the development of ABC based on a population of ethnic Kazakh women. Using the odds ratio approach, we tried to quantify the strength of an association between haplotypes of genetic markers located in exons (proteins take part in physical interactions) and eBC.

## 2. Results

### Genome-Wide SNP Genotyping Results

Genotyping of 654,027 autosomal genetic markers revealed 25,323 SNP variants with high no-call rates and three samples with a genotype call rate less than 0.95. They were filtered out and excluded from the analysis. Additionally, 231,362 variants with an MAF of less than 3% were filtered out from the genotyping data screened. A futher 70,304 markers did not correspond to the HWE and were also excluded from the analysis. In total, 327,038 SNP markers were tested for the association analysis.

The associative analysis of the data was carried out using a multiplicative model of inheritance (*A* allele vs. *B* allele). Totally, 18,508 SNPs showed OR associations with a classical confidence level not exceeding a 5% error rate (*p* = 0.05). According to the analysis, there were no genetic markers achieving a GWAS significance threshold *p*-value of <5 × 10^−8^ and internal Bonferroni correction *p* = 1.5 × 10^−7^. Visualization of the obtained OR results were plotted by a Manhattan plot, –log10 (*p*-value) against the chromosome position (Figure 1).

The most significant genetic markers identified were genetic loci, and SNPs reached the threshold of 5 × 10^−6^ (−log_10_P = 5.3) on the first, second, third and sixth chromosomes. A genetic locus on the first chromosome included four SNPs in high linkage disequilibrium (R^2^ = 1, D’ = 1, *p* < 0.0001) within a gene sequence (intron), and a genetic locus on the second chromosome included three SNPs in linkage disequilibrium (R^2^ = 1, D’ = 1, *p* < 0.0001) within a gene sequence (intron). The other three genetic variations on the first, third and sixth chromosomes were in linkage equilibrium with neighboring markers and located within poorly characterized intergenic sequences. Data for these markers are presented in a summary Table 1.

Since statistical methods for calculating OR and *p*-value are largely dependent on the sample size, and taking into account the sample size of this study, we took a comprehensive approach to assessing the role of the analyzed genetic markers.

Since nucleotide variations in exonic regions of genes can potentially affect the structure of the protein they encode and, thereby, protein–protein interactions (PPI), we analyzed genes whose exons contain associated SNPs (*p* < 0.05). A total of 909 polymorphisms were selected within the exons of 757 genes. To identify direct physical interactions between the proteins of the analyzed genes, we projected them into a PPI network created by the web-based STRING database [12]. The analysis was carried out taking into account that the interaction of the two proteins is a part of the physical complex and not the functional status in cells and their co-expression. A general network was created with possible PP physical interactions of 716 nodes, 358 edges and the network enrichment of 4.37 × 10^−7^. Physical PP interactions with a minimum required interaction score of 0.700 (high confidence) were noted for 132 PP pairs.

Clustering of the physical PPI network revealed the main sub-networks: SRC-associated (ten nodes, eight edges, *p* = 5.79 × 10^−8^) and ERCC6-associated (eight nodes, eleven edges and *p* = 3.02 × 10^−12^). Also, there were five small sub-networks consisting of several nodes and edges with significant enrichment *p*-values (Figure 2).

Further, 132 physical PPI pairs were analyzed for associations between corresponding haplotypes and eBC. Polymorphisms located within exons of the selected genes pairs were accepted for this analysis. Out of 132 pairs, 52 pairs were identified.

The associative analysis was carried out using a standard approach of odds ratios (OR) by combining selected earlier polymorphisms of genes into corresponding haplotypes. In Figure 3, four types of possible haplotype (nine haplotypes) combinations are presented. Due to our limited case/control sample size, it was difficult to catch all the possible haplotypes to calculate statistically significant OR results. We simplified these possible haplotypes to groups according to the numbers of negative and positive alleles in each haplotype for the case and control (Figure 3). A multiplicative model of inheritance based on the assumption that penetrance depends on the number of copies of the predisposing alleles. We used it as the main one to establish correlations. Results for this model are presented in Appendix A. The simplified haplotype groups were used to estimate the additive model where the penetrance value of Group II (compound heterozygotes) lies between the penetrance values for both Group I and III (with the prevalence of negative (Group I) or positive (Group III) alleles in haplotypes) (Appendix A).

According to the multiplicative model, seven PPI genes pairs achieved an internal Bonferroni correction threshold (*p* = 9.4 × 10^−4^). Table 2 shows the association results for these pairs. The most significant association was for negative alleles of *A2M*/*LRP1* gene pair (OR = 2.15; *p* = 7.7 × 10^−6^). The combination of haplotypes in the additive model for *A2M*/*LRP1* gene pair confirmed these findings and showed a strong negative effect for Group I (OR = 2.98; *p* = 3.3 × 10^−4^). This was strengthened by the presence of the second signal from the *A2ML1*/*LRP1* gene pair. The negative and positive allele distribution for rs1860967/rs1800137 gave OR = 1.65 and *p* = 2.8 × 10^−4^. These results were confirmed by the additive model (OR = 3.10, *p* = 4.6 × 10^−4^).

Another analyzed gene *HEATR1* had three polymorphisms for the combination analysis with polymorphisms of genes *NOP14* (rs2515960) and *NOL10* (rs3732111). Out of six pair combinations, only one *HEATR1/NOP14* pair (rs60920266/rs2515960) was confirmed by both multiplicative and additive models. According to the multiplicative model, the distribution of negative and positive alleles in the case and control groups were statistically different (OR = 1.58; *p* = 9.1 × 10^−4^). These findings were confirmed by the appropriate distribution of haplotypes in the additive model, which showed a high OR = 2.94 for Group I and statistically significant *p* = 8.2 × 10^−4^.

Polymorphism rs2515960 in the *NOP14* gene also had a statistically significant association in combination with *NOL10* gene polymorphism (rs3732111). The multiplicative model showed an elevated level of OR = 1.52 (*p* = 7.3 × 10^−4^) and was confirmed by the additive model (OR = 2.07), but with a decreased level of confidence (*p* = 7.6 × 10^−3^). The other polymorphism combinations in the *HEATR1, NOP14,* and *NOL10* genes had not reached the Bonferroni significance level.

Also, elevated OR results with decreasing confidence levels were shown for two genes pairs, *SLX4/TOPBP1* and *GABRP/NSF.* The gene pair *SLX4/TOPBP1* had significant OR = 1.73, *p* = 7.2 × 10^−4^ in the multiplicative model but high OR = 3.71 and lower *p* = 7.2 × 10^−3^ in the additive model. The pair of *GABRP/NSF* had OR almost twice as high (OR = 1.93, *p* = 0.00033) in the multiplicative model but a high OR and lower significance of genotype distribution in the additive model (OR = 5.74, *p* = 0.0018).

The pair of *ACKR1/CD82* had the opposite results of OR = 1.44 and *p* = 3 × 10^−3^ in the multiplicative model vs. OR = 3.98 for the Group I and high *p* = 4.7 × 10^−5^ in the additive model.

Thus, the genome-wide association analysis revealed that out of 327,038 SNP markers, 18,508 SNPs had OR associations with the classical confidence level of *p* ≤ 0.05. However, none of these markers reached the GWAS or internal Bonferroni correction thresholds. The most significant markers were noted for intron polymorphisms within the *CHI3L2* and *MGAT5* genes. Further analysis of exon polymorphisms in haplotypes revealed that among 132 PP pairs, five exhibited statistically significant OR values for corresponding haplotype combinations. The other polymorphism combinations in the PP pairs did not reach the Bonferroni internal significance level, but are still of interest for further research.

## 3. Discussion

Generally, in this study we followed the hypothesis that cancer is a complex multi-stage process of enrichment for normal molecular–biochemical (cell–cell communication, division, growth, etc.) processes with negative «elements» predisposing to a shift towards cancer-related processes and pathways. The enrichment degree of normal processes apparently determines the approximation degree to the point of no return (hotspot) and further malignancy (malignancy hotspot). The enrichment and approximation degree may be determined by a combination of known (age, gender, mutations, radiation, lifestyle etc.) and unknown factors. They are parts of more general processes: environment–genetics–function (Figure 4). Each of the three interacting processes is huge and complex due to the large number of components involved. They are inseparable from each other as they constitute life. Probability assessment of cancer outcome depends on the number of these factors taken into account.

This picture represents different combinations of three main processes to produce a combination risk of the disease: Environmental, Genetics and Function. We consider that each of the processes include a huge number of elements. Not all elements within one process correspond to the disease cause of the structured reactome interactions.

Understanding genetic processes is an important step in cancer susceptibility and development. The identification and characterization of specific genes that play a role in cancer development, as well as the evaluation of genetic variants that may contribute to individual susceptibility to cancer, provide insights into the genetic processes underlying different types of cancer. Moreover, they may partially influence the functional processes from cells to the whole body.

In this regard, we tried to clarify whether there are genetic markers involved in eBC development in Kazakh women by associative study. To investigate the involvement of common genetic variants (SNPs) associated with eBC in the population of Kazakh women, we performed genome-wide association analysis using a total of 144 breast cancer cases and 163 control individuals. The main steps of the analysis are presented on the scheme (Figure 5).

Selection of cases and controls should be conducted carefully to minimize bias and ensure accurate population representation. To create breast cancer case and control groups, we used three main population characteristics: ethnicity, gender and age.

Allele frequencies may be different among different populations contributing to heterogeneity in disease burden. In our study, only women of Kazakh ethnicity from one geographic zone (long-term residents, Almaty city, Kazakhstan) were chosen. A total of 16% of participants in the case group were not from Almaty city but from nearby areas of the city. By choosing participants for the case and control groups from the same geographic region, we tried to minimize sub-ethnic factors. Additionally, participants from the same geographical region are exposed to the same environmental factors. This reduces the risks associated with environmental factors. The case/control groups matched each other by age structure. Age is an important factor in «case-control» studies, as it can influence the disease’s outcome and exposure. Clarifying this, BC is an age-dependent process, the main debut of which occurs over 60 years of age. The development of eBC gives reason to believe that genetic processes are more enriched in “negative” elements than functional and environmental processes. For example, 5–15% of eBC cases are caused by rare pathogenic mutations in the highly penetrant *BRCA1/2* genes. Therefore, the enrichment of genetic processes with highly penetrant alleles significantly increases the likelihood of a cancer outcome in a short time. It is similar to believe that the remaining cases of eBC are due to the enrichment of genetic processes with a larger number of medium and/or low penetrant genetic variations [13], thus increasing the time interval before cancer manifestation. The development of BC in older age groups may indicate that along with the enrichment of genetic processes, functional and environmental processes become highly enriched. It should also be assumed that individuals may have a level of enrichment of genetic processes closer to the average but a higher enrichment of functional and environmental processes.

Here, we have the ambiguous problem of selecting a control group considering age for cases of eBC. There is a possibility that in the control group of young women, there may be individuals who will develop cancer in the future. Since many of them may be carriers of medium- and low-penetrance alleles, the problem of their influence on the statistical data of association analysis remains an open question. In our study, we selected control participants with a predominance of older age subgroups of 36–40 and 41–45 years. In addition, we did not include participants with a family history of BC in the experimental group, thereby filtering out cases with highly penetrant alleles. An association analysis considering an additional older age group will probably contribute to a more comprehensive GWA analysis and obtain comprehensive results.

Our case-control associative analysis revealed two loci with high LD. We compared the revealed data with LD patterns in 1000 G populations by using the online NCBI LDpop tool [14]. Locus1 for the healthy Kazakh women group showed complete LD for all four SNPs in the locus (R2 = 1, D’ = 1; *p* < 0.0001). These data were consistent with all populations available (32 populations) in the NCBI database. Locus2 included three SNPs and only two showed 100% LD: rs2289467 and rs35237563. The obtained data were consistent with 27 out of 32 populations compared to the NCBI LDpop base. Four populations of China CHB, CHS, CDX, KHV and one of the general East Asian (EAS) population had incomplete LD (R2 = 0.7–0.79, *p* < 0.0001) for these two polymorphisms. The third (3d) SNP (rs3828184) from the studied Locus2 had incomplete LD (80%) with the first two SNPs (rs2289467 and rs35237563). LD for rs3828184/rs2289467 and rs3828184/rs35237563 had similar results of R2 = 0.8, *p* < 0.0001 for our population. LD for rs3828184/rs2289467 matched four ethnically different populations out of 32 populations: BEB (Bangladesh), JPT (Japan), CLM (Colombia), and one general AMR (Ad Mixed American) population. LD for rs3828184/rs35237563 matched the same four ethnically different populations plus another four populations: CHB (China), PJL (Pakistan), KHV (Vietnam), and the general SAS (South Asian) population. Known factors such as genetic drift, natural selection, recombination, and migrations may cause observed differences between populations.

Our GWA analysis did not show significance values above the threshold level for GWAS studies, nor even Bonferroni internal significance. Considering the sample size of our groups, obtained OR results for single SNPs of two Loci1 and two with *p*-value < 0.000001 are important candidates for the genetic markers. These SNPs lie within the introns of genes *CHI3L2* (Locus1) and *MGAT5* (Locus2). The *CHI3L2* and *MGAT5* genes are located on the first and second chromosomes, respectively. Both genes are secretory proteins and involved in processes of outside-cell communications through oligosaccharide modifications. CHI3L2 is a chitinase-like protein binding to poorly understood oligosaccharides in mammals [15]. This protein is expressed in chondrocytes of cartilage [16], the gastrointestinal tract and in a subset of cortical neurons. Over expression was involved in different pathologies: osteoarthritis [17], amyotrophic lateral sclerosis [18], multiple sclerosis and cancers [19,20]. Prognostic significance has been noted for renal cancer [21] and glioma [22,23]. A recent study by Ling Xue et al. proposed a prognostic role in gastric adenocarcinoma [24]. CHI3L2 is mainly secreted by tumor-associated macrophages and associated with metastasis progression and poor outcome [23,24]. The second gene, MGAT5, expresses a glycosyltransferase family member catalyzed by an essential step in the biosynthesis of branched, complex-type N-glycans (oligosaccharides). These branched glycans attach to the cell’s surface and, thus, change surface levels of protein- and lipid-bound oligosaccharides [25,26]. Alterations in branched glycan spectrum on the cell surface are correlated with significant changes in normal adhesion, migration and proliferation processes [25,26]. MGAT5 protein is expressed in many types of cells, and over-expression was shown in different malignancies. Loss of function was noted for tumor suppression and inhibition of metastasis on model objects and humans [27,28,29,30]. Thus, both proteins are involved in mechanisms of normal cell–cell communications, adhesion and sells quorum through the cell surface oligosaccharide complex’s metabolism. Modifications in the oligosaccharide complexes’ structure, their bioavailability on cell surface and functional activity are associated with tissue microenvironment disruption. These conditions are predisposing factors for carcinogenesis.

Although, intronic sequences are poorly understood, there is evidence demonstrating mechanisms of intronic variants cause diseases and cancer. Variations in the canonical splice sequences, deep intronic variants, and intron retention variants may cause abnormal splicing [31,32]. They may account for 15–60% of human diseases [33]. Nowadays, identifying disease-associated splicing associated variants has become more important than ever. In this aspect, polymorphisms found within the genes discussed in our study are of great interest for future research. Additional research using other methods and approaches is required.

Polymorphisms in exonic regions have more specific effects on the structural features of proteins, protein complexes or mRNAs. In this study, we analyzed associations of combinations for all exonic polymorphisms showed earlier OR > 1 results (*p* < 0.05). Combinations were made on the basis that protein–protein interactions are part of a physical complex (STRING database tool). Thus, the obtained PP pairs had their SNP combinations and corresponding haplotypes for further associative analysis by the odds ratio approach. The results showed significant negative and positive allele distribution differences in case vs. control groups observed for seven pairs. The appropriate genotype distribution of the additive model confirmed these results. One pair of SNPs in the *ACKR1/CD82* genes (OR = 1.44, *p* = 3 × 10^−3^) was included because it had a high level of significance in genotypes distribution between case and control groups observed in the additive model (OR = 3.98, *p* = 4.7 × 10^−5^ for Group I). These differences may be explained by the unequal contribution of the studied alleles in the corresponding genotypes.

Results showed the highest double signal from the *A2M/LRP1* genes pair. This combination is remarkable in that the high *A2M* gene expression was noted for the naked mole-rat, a subterranean rodent known for its resistance to hypoxia, hypercapnia, aging and cancer [34]. It was shown that LRP1 is a well-known receptor for blood A2M protein [35,36]. Through interaction with the LRP1 receptor, the A2M protein mediates a clearance of different proteins from cells and modulates signaling pathways [37]. Notably, the A2M expression decreases with age [38]. In vitro and in vivo experiments on model objects show tumor suppression functions for the A2M protein. Notably, the A2M protein inhibits tumors independent of their origin [36].

In our study SNPs of the *A2M* and *LRP1* genes are a good example of low penetrance polymorphisms. Separately, polymorphism rs669 of the *A2M* gene showed OR = 2 with *p* = 1.9 × 10^−3^ and rs1800137 of the *LRP1* gene OR = 2.4 with *p* = 1 × 10^−3^. Together, these polymorphisms give more predisposing haplotype in corresponding individuals. It should be noted that polymorphisms in the *A2M* and *LRP1* genes belong to the different types of substitutions: the A2M carries a missense variation, and *LRP1* carries a synonymous variation. Protein interactions may be due to physical contact or bioavailability of one of the proteins. In our case, the last one may be due to synonymous substitution in mRNA structure and its bioavailability for translation into complete protein LRP1. Maybe this process is tissue-specific. The frequency of interactions may determine the efficiency of cell signaling. Interestingly, a protein with similar functions and high homologous structure, A2ML1, also showed statistically significant results in *A2ML1/LRP1* genes pair (OR = 1.65, *p* = 2.8 × 10^−4^). The LRP1 protein is the ligand for this protein [39]. Thus, our results indicate that haplotypes carried negative variants of polymorphisms rs669/rs1800137 in the *A2M/LRP1* gene pair have an elevated OR to eBC. It also highlights the importance of signaling pathways mediated by this interaction and is of great interest for future research.

A dual signal was observed for the rRNA processing, implicating the *HEATR1/NOP14* and *NOP14/NOL10* in the positive regulation of rRNA processing and transcription by RNA polymerase I. Pre-rRNA processing is tightly regulated, involving many cellular components acting alone or as part of a complex. Studied PP pairs belong to the nucleolar processome complex. The *HEATR1/NOP14* pair includes two missense variants in fifteen and eight exons, respectively. These genetic variations are probably important for target interactions of the HEATR1 and NOP14 proteins with each other or rRNA. It was established that cancer cells have upregulated activity of rRNA biogenesis and abnormal rRNA modifications. rRNA biogenesis is complex, and there is evidence that it is directly associated with carcinogenesis [40].

## 4. Conclusions

The contribution of genetic markers to disease predisposition is complex and depends on various factors ranging from genetic variation penetrance to statistical methods applied. The odds ratio method reflects the strength of the link between the two traits studied. The link strength may depend on the penetrance of the genetic marker and may be eliminated if the penetrance is low. Low penetrant polymorphisms do not disrupt the gene’s protein or RNA, but they modify and may be more common in the population. Secondly, genetic variation cannot functionally exist on its own, only in interactions with molecular targets. These interactions could enhance or lower the link strength, making a complex phenotype and reducing statistical association data. In this study, we tried to evaluate not only the contribution of SNPs to the susceptibility of eBC, but also combinations of SNPs for assessment of significant protein–protein pairs in eBC. The findings underscore the significance of incorporating medium- and low-penetrant SNPs in combination analyses, as they enable the identification of crucial interactions contributing to the disease.

It is important to note that, given the limited scope of our study involving patients from Almaty and the Almaty region, the findings may not accurately depict the actual situation for the entire population of Kazakh women. Therefore, for future research, we plan to consolidate existing databases and include more patients to obtain a more representative cohort. This will help identify population-specific SNP markers and breast cancer-associated genetic variants in ethnic Kazakh women applicable in clinical practice. Implementing them into national screening programs will contribute to reducing breast cancer incidence in Kazakhstan.

## 5. Materials and Methods

### 5.1. Study Subject

The study was conducted at the Kazakh Research Institute of Oncology and Radiology and the Institute of Genetics and Physiology in Almaty, Kazakhstan. A total of 330 unrelated Kazakh women were recruited in the «case-control» study. After all data quality control verification steps, a total of 307 participants were selected for further statistical analysis. The case group comprised 144 unrelated female patients of Kazakh ethnicity diagnosed with eBC aged between 19 and 40 years at the time of diagnosis. Accurate diagnoses were determined by qualified oncologists using mammography and histopathology. Exclusion criteria comprised other cancers and any other early-onset and inherited inflammatory, cardiac, hepatic, and neurological diseases. A total of 60% of patients were long-term residents of Almaty City, and 40% were from the nearby areas of Almaty City.

A control group included 163 healthy individuals without BC diagnosis, any family history of BC or other cancers, or cardiac, hereditary, inflammatory, and autoimmune diseases. Participants in the control group were selected according to the age of the case group. All personalized data of the participants (case and control groups) were enrolled in questionnaires that included demographic status and habits (tobacco, alcohol, and diet). A total of 84% of individuals were long-term residents of Almaty City, and 16% were from the nearby areas of Almaty City.

Biomaterials were obtained after signing a voluntary informed consent to participate in the research. This research work was approved by the Ethics Committee of the Kazakh Institute of Oncology and Radiology (No. 04 from 23 September 2020) and adhered to the principles of the Declaration of Helsinki.

### 5.2. Genomic DNA Isolation and Genotyping

Genomic DNA was isolated from EDTA-treated peripheral blood samples using the Qiagen DNA blood mini kit (Qiagen, Hilden, Germany). Qualitative and quantitative characteristics of the DNA samples were estimated by spectrophotometry (NanoDrop, Thermo Scientific, Waltham, MA, USA). Isolated DNA samples were stored at −20 °C.

Scanning for known genetic markers was performed through microarray-based SNP genotyping using the Infinium Global Screening Array (GSA-24 v3.0, Illumina, San Diego, CA, USA). A method-specific workflow was followed according to the manufacturer’s protocol. The normalized data was extracted and processed using the GenomeStudio Genotyping Module v2.0 software. The quality of the samples was assessed by the frequency of genotype calls using the Call Rate > 0.95 algorithm. The GenTrain score was used to characterize the data clustering algorithm (>0.7) qualitatively. The data quality control for subsequent statistical analysis was carried out in accordance with the following criteria: (1) exclusion of SNPs and samples with >1% of data failed genotyping, (2) exclusion of SNPs with a minor allele frequency less than 0.03 (MAF < 3%), (3) exclusion of SNPs with *p* < 0.05 for Hardy–Weinberg equilibrium (HWE) and (4) exclusion of SNPs on X-chromosomes.

The data processing and clustering of all samples was performed in duplicate and scored by different individuals.

### 5.3. Statistical Analysis

Values characterizing the case and control populations were calculated in percentage and were expressed as mean ± standard deviation or standard error values. The Student *t*-test was used to compare the distribution of variables between case and control cohorts. A value of less than 0.05 was accepted as significant.

The expected genotype frequencies for case and control groups were calculated in accordance with standard Hardy–Weinberg equilibrium using the conventional Pearson’s chi-square test (χ2). A probability value (*p*-value) of 0.05 was used to evaluate the significance of a Chi-Square test.

Estimation of a statistical relationship between the sign of the disease and the genetic marker was calculated according to basic statistical analysis in genetic case-control studies [41]. A multiplicative model was used for disease penetrance assessment that imply a specific relationship between genotype and phenotype [42]. Estimation of the coefficient of relative risk was calculated by the method of “odds ratio” (OR) in conjunction with an estimate 95% confidence interval (95% CI) and the “Chi-square” test (χ2) for the degrees of freedom = 1. Generally accepted threshold for GWAS studies *p* = 5 × 10^−8^ and internal Bonferroni correction *p* = 1.5 × 10^−7^ were used as an adjustment for multiple hypothesis testing.

Annotation of identified significant polymorphisms was performed using the ClinVar, 1000 Genomes, ExAC, Cosmic, and dbSNP databases. A theoretical assessment of the possible impact of non-synonymous SNP substitutions was carried out using the PolyPhen-2 program.

For the reconstruction of possible protein–protein interactions, a STRING database (version 12.0) was used (https://string-db.org/cgi/input?sessionId=bOgLukFsBdI0&input_page_active_form=multiple_identifiers, accessed on 14 August 2023). The STRING database is a web-based tool for creation and visualization of complex biological networks. A direct or physical subnetwork association algorithm was applied in the obtained PPI network analysis, which constructs PPI edges based on known physical complexes, not functional and co-expression interactions. These interactions were created based on the known experimental and/or biochemical data of human and other organisms and from data in curated databases or co-mention in Pubmed abstracts database. A minimum required interaction score of 0.700 was applied. PPI enrichment *p*-value < 0.05 was considered as statistically significant.

## Figures and Tables

**Figure 1 genes-15-00108-f001:**
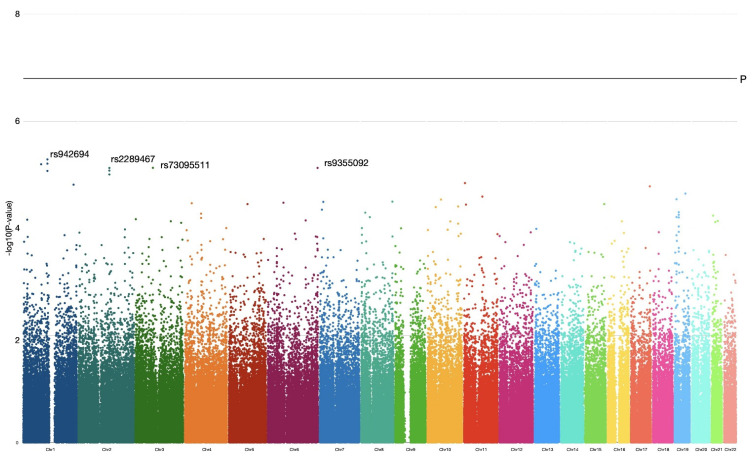
Manhattan plot showing the level of statistical significance of detected associations for each analyzed polymorphism. The *x*-axis includes the genomic coordinates of analyzed polymorphisms (SNPs) for each chromosome. The highest *p* value (OR = 2.51; *p* = 5.2 × 10^−6^) was observed for rs942694 in the first chromosome. The horizontal black line shows the GWAS threshold at *p* = 5 × 10^−8^ (−log_10_P = 7.3). Chr: chromosome.

**Figure 2 genes-15-00108-f002:**
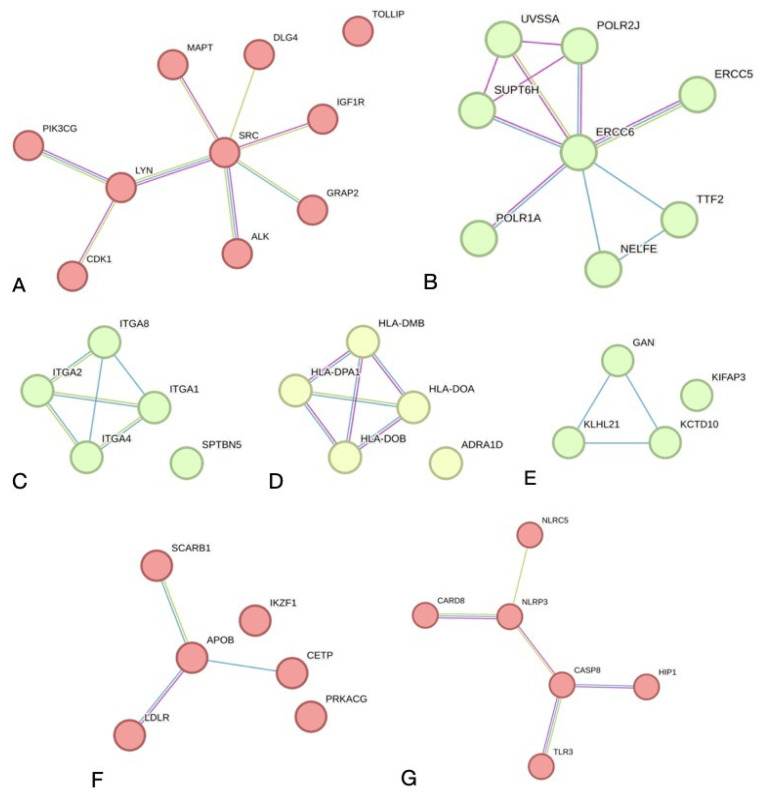
PPI networks based on the STRING on-line tool. (**A**)-SRC-kinase activity network (10 nodes; 8 edges; *p* = 5.79 × 10^−8^); (**B**)-ERCC6-excision repair PPI network (8 nodes; 11 edges; *p* = 3.02 × 10^−12^); (**C**)-integrin cell adhesion PPI network (6 nodes; 3 edges; *p* = 1.43 × 10^−5^); (**D**)-HLA Class II PPI network (5 nodes; 6 edges; *p* = 2.22 × 10^−15^); (**E**)-Ubiquitin ligase complex PPI network (4 nodes; 3 edges; *p* = 2.61 × 10^−6^); (**F**)-Lipoprotein regulation network (6 nodes; 3 edges; *p* = 1.43 × 10^−5^); (**G**)-Pyroptosis PPI network (6 nodes; 5 edges; *p* = 2.48 × 10^−8^).

**Figure 3 genes-15-00108-f003:**
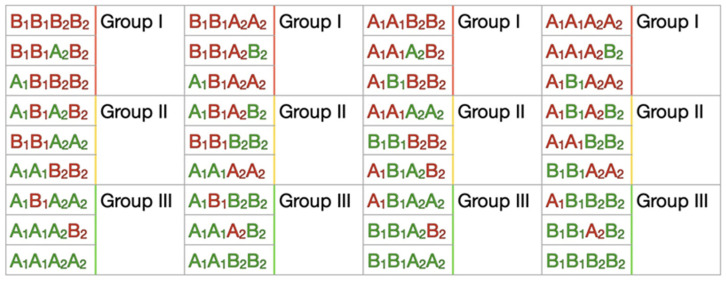
Types of possible haplotypes and their combinations. Red—alleles with negative effect; green—alleles with protective effect (positive). Group I contain three haplotypes with prevalence of negative alleles; Group II contains three equal proportion of negative and positive alleles; Group III contains three haplotypes with prevalence of positive alleles.

**Figure 4 genes-15-00108-f004:**
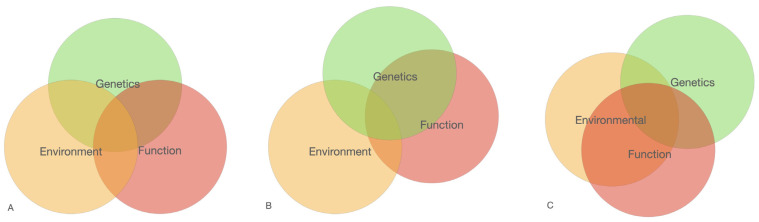
Scheme of the main processes involved in cancer predisposing. The diagrams reflect the proposed relationships between the three main factors involved in the development of cancer. The diagram (**A**) shows an equal factors contribution; (**B**) shows the prevalence of Genetics and Function factors vs. Environment (for example, rare negative mutations); (**C**) shows the prevalence impact of Environmental and Function factors (for example, age-dependent burden of negative elements in exposure of unfavorable Environmental factors).

**Figure 5 genes-15-00108-f005:**
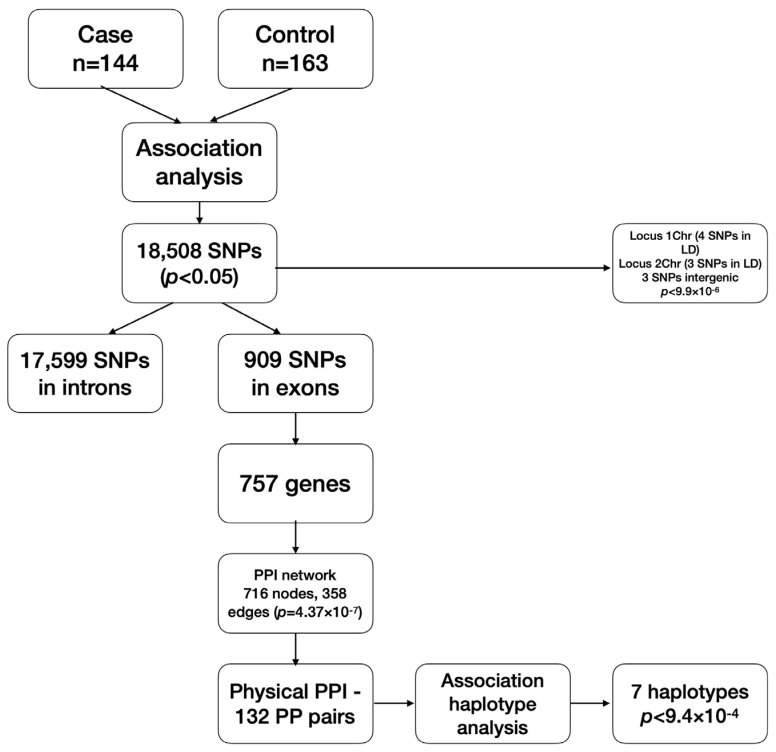
Scheme of the main steps conducted in the research. The scheme represents sequential steps of two-stage association analysis and obtained results. SNPs—single nucleotide polymorphisms; PPI—protein-protein interactions; LD—linkage disequilibrium.

**Table 1 genes-15-00108-t001:** Significant markers on chromosomes analyzed (threshold of 5 × 10^−6^).

SNP ID	Base Change	Chr	Position	MAF (Database)	MAF Case/Control	Effect Allele	OR (95%CI)	*p*-Value	Gene
SNP 1									
rs55914748	T/G	1	82884721	0.22 (G)	0.23/0.09	T	0.36 (0.22–0.56)	6.4 × 10^−6^	-
						G	2.81 (1.77–4.46)		
Locus 1									
rs942694	T/C	1	111784138	0.19 (C)	0.29/0.14	T	0.4 (0.27–0.6)	5.2 × 10^−6^	*CHI3L2*
						C	2.51 (1.68–3.75)		
rs942693	T/C	1	111784158	0.19 (C)	0.29/0.14	T	0.40 (0.27–0.7)	6.2 × 10^−6^	*CHI3L2*
						C	2.49 (1.66–3.72)		
rs5003370	T/G	1	111784509	0.19 (G)	0.29/0.14	T	0.4 (0.27–0.6)	5.2 × 10^−6^	*CHI3L2*
						G	2.51 (1.68–3.75)		
rs5003373	A/G	1	111784517	0.19 (G)	0.29/0.14	A	0.41 (0.27–0.61)	8.5 × 10^−6^	*CHI3L2*
						G	2.44 (1.64–3.65)		
Locus 2									
rs2289467	T/C	2	135180550	0.08 (T)	0.15/0.05	T	3.71 (2.02–6.83)	8.4 × 10^−6^	*MGAT5*
						C	0.27 (0.15–0.5)		
rs35237563	A/G	2	135182574	0.08 (A)	0.15/0.05	A	3.71 (2.02–6.83)	8.4 × 10^−6^	*MGAT5*
						G	0.27 (0.15–0.5)		
rs3828184	T/G	2	135200567	0.09 (G)	0.14/0.04	G	3.8 (2.03–7.13)	9.9 × 10^−6^	*MGAT5*
						T	0.26 (0.14–0.5)		
SNP2									
rs73095511	T/C	3	72385521	0.16 (C)	0.1/0.23	T	2.82 (1.77–4.5)	7.4 × 10^−6^	-
						C	0.35 (0.22–0.56)		
SNP3									
rs9355092	T/G	6	169210922	0.3 (G)	0.37/0.21	T	0.44 (0.31–0.63)	7.5 × 10^−6^	-
						G	2.26 (1.58–3.24)		

**Table 2 genes-15-00108-t002:** Odds ratios (ORs) and 95% confidence intervals (CIs) for significant SNP pairs.

No	Alleles		Protein-Protein Pair	Case		Control		OR (95% CI)	*p*-Value
	Negative	Positive		Negative Alleles	Positive Alleles	Negative Alleles	Positive Alleles		
1	G	A	A2M__rs669_	103	473	60	592	2.15 (1.53–3.02)	0.000007
	T	C	LRP1__rs1800137_						
2	C	T	A2ML1__1860967_	156	420	120	532	1.65 (1.26–2.16)	0.00028
	T	C	LRP1__rs1800137_						
3	C	T	HEATR1__rs60920266_	469	107	479	173	1.58 (1.21–2.08)	0.00091
	A	G	NOP14__rs2515960_						
4	A	G	NOL10__rs3732111_	418	158	413	237	1.52 (1.19–1.94)	0.00073
	A	G	NOP14__rs2515960_						
5	G	A	SLX4__rs3810813_	507	69	528	124	1.73 (1.25–2.37)	0.00072
	T	C	TOPBP1__rs10935070_						
6	C	A	GABRP__rs1063310_	84	492	53	599	1.93 (1.34–2.78)	0.00033
	A	G	NSF__rs199533_						
7	A	G	ACKR1__rs12075_	216	360	181	435	1.44 (1.13–1.84)	0.003
	A	G	CD82__rs1139971_						

## Data Availability

All data generated or analyzed during this study are available from the corresponding author upon request.

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
