# Peer review of "Association of Genetic Markers with the Risk of Early-Onset Breast Cancer in Kazakh Women"

_genes, 2024, doi:10.3390/genes15010108_

Round 1
Reviewer 1 Report
Comments and Suggestions for Authors
The article entitled “Association of Genetic Markers with the Risk of Early-Onset Breast Cancer in Kazakh Women” is well-written and, from my point of view, authors have made a great job preparing an interesting manuscript for Genes. In spite of this and before its publication, certain issues should be improved. Please see below my list of comments:
- In the introduction, at the end, a detailed description of the layout of the manuscript should be included.
- In results: “study subjects” it is not part of results if not of materials and methods.
- Line 86: it is said “60% of patients were from the same region of residence”. What does it means? Please, explain.
- Line 88-89: it is said “16% of control individuals were from the same region of residence”. What does it means? Where does the rest of individuals come from? Please explain.
- In line 126 it is said: “we applied the STRING database” but according to what it can be seem it must me a methodology linked to graph theory. Please, explain in more detail. Please note that was indicated in the materials and method section is not enough.
Reviewer 2 Report
Comments and Suggestions for Authors
The article on the association analysis of genetic markers with early-onset breast cancer (eBC) in Kazakh women presents several potential incongruences and areas for improvement.
The abstract mentions two risk intronic loci associated with the MGAT5 and CHI3L2 genes. The biological relevance of these loci and their potential influence on the onset of breast cancer, however, are not sufficiently explained. The discovery of genetic risk factors is given further importance in conclusion, highlighting the impact of medium and low-penetrant SNPs.
In the introduction, breast cancer is identified as a serious health issue in Kazakhstan, even though potential biases in the research population are not addressed. The report should note the limitations of the study about population variation and the potential for the results to not apply to all Kazakh women.
Moreover, the publication lacks precise information about any confounding variables that can affect the study's results, such as lifestyle and environmental factors. Expanding on these details would improve the validity of the study.
Because of the extensive yet intricate results section, it might be difficult for readers to grasp the approach and appreciate the relevance of the findings. The data could be presented more simply in this paper, and further information about the functional implications of the found genetic markers should be included.
The need for more research is briefly mentioned in the article's conclusion, but it would be beneficial to include a more detailed discussion of possible future paths, such as investigating the functional implications of the identified genetic markers and applying research findings to clinical settings.
Round 2
Reviewer 2 Report
Comments and Suggestions for Authors
The recommendations have led to the proposed changes being made to the article. The suggestions made resulted in improvements in the introduction, while the conclusion section discusses the study's weaknesses. Further adjustments, in my opinion, are not necessary.